# LOAd703-induced tumor microenvironment gene engineering in combination with atezolizumab in metastatic malignant melanoma: a phase I/II trial

O. Hamid [1,8], V. Ekström-Rydén [2,3,8], I. Mehmi [1], D. Wang [4,7], M. Patel [4], S. Alsaqal [5], S. Irenaeus [2,3], C. Nordström [6], LC Sandin [6], H. Grauers Wiktorin [2], T. Lövgren [2], E. Eriksson [2,6], J. Leja-Jarblad [6], A. Loskog [2,6,9] ✉ & GJ Ullenhag [2,3,9]

This phase I/II study evaluated the capacity of tumor microenvironment (TME) gene engineering by intratumoral injection of a viral vector encoding a designed CD40L and 4-1BBL combined with intravenous atezolizumab (anti-PD-L1 antibody) to induce immune activation in 24 patients with stage IV malignant melanoma refractory to PD-1 inhibition. Primary objective was tolerability. Tumor response, pharmacokinetics, and biomarker evaluations were secondary. Treatment was well-tolerated in both dose cohorts ($1\times10^{11}$ and $5\times10^{11}$ viral particles). Th1 immune biomarkers was increased in the TME (NanoString) as well as in blood (Olink). In long-term survivors, we observed increased markers for T cell fitness and for the immunoproteasome. The overall response rate was 17% accordingly to RECIST 1.1 and disease control was noted in 54%. Forty-six percent of patients were still alive two years post enrollment. The lower dose showed very encouraging results with a median progression-free survival of 9.7 months and median overall survival of 26.3 months (post-hoc analyses). In conclusion, TME gene engineering may have re-sensitized refractory patients to checkpoint treatment or acted alone to control tumor growth. The small sample size and single arm design limits effect interpretation but the data shows promise for continued clinical investigation. Study registration: NCT04123470.

Cutaneous malignant melanoma (CMM) is the fifth most common form of cancer in the Western world, with a tripled incidence in the US over the last decades[1]. Immune checkpoint inhibition (ICI) targeting PD-1, CTLA-4, and LAG-3 has revolutionized advanced CMM treatment, increasing the 5-year survival rate to 50% and is standard-of-care together with protein kinase inhibitors for BRAFV600E-mutated CMM[2–4]. Despite these breakthroughs, therapy resistance is common[5]. Resistance depends on low numbers of tumor-infiltrating T

[1]The Angeles Clinic and Research Institute, A Cedars Sinai Affiliate, Medical Oncology, Los Angeles, CA, USA. [2]Department of Immunology, Genetics and Pathology, Science for Life Laboratory, Uppsala University, Uppsala, Sweden. [3]Department of Oncology, Uppsala University Hospital, Uppsala, Sweden. [4]Baylor College of Medicine, McNair Campus, Houston, TX, USA. [5]Department of Surgical Sciences, Radiology & Molecular Imaging, Uppsala University, Uppsala, Sweden. [6]Lokon Pharma AB, Uppsala, Sweden. [7]Present address: Department of Internal Medicine at UT Southwestern Medical Center and the Division of Hematology and Oncology, Dallas, TX, USA. [8]These authors contributed equally: O. Hamid, V. Ekström-Rydén. [9]These authors jointly supervised this work: A. Loskog, GJ Ullenhag. ✉e-mail: angelica.loskog@igp.uu.se

cells and their inability to kill tumor cells. An approach for ICI-resistant patients is to reprogram the tumor microenvironment (TME) to sensitize the patients to ICI and to promote in vivo generation of T cells. The TME consists of multiple cell types, including fibroblasts, vascular endothelium, immune cells, and products released by them that collectively induce immunosuppression[6,7,8] and reduce activation of a type 1 T helper cell (Th1) response needed to combat the tumor[9].

LOAd703 is a TME gene engineering adenoviral vector with oncolytic capacity, transferring Th1-type immunostimulatory genes to the TME[10–12]. While it can infect and induce expression of the genes in any infected cell, the replication-induced oncolysis occurs only in tumor cells. LOAd703 encodes trimerized, membrane-bound CD40L (TMZ-CD40L) and 4-1BBL. CD40L is an important initiator of immune responses by inducing maturation of DCs, M1 macrophage differentiation, chemokine and cytokine release from stroma cells, and upregulating attachment receptors on endothelial cells[13–15]. Importantly, LOAd703-stimulated DCs upregulate CCR7[16,17], which promotes their translocation to lymph nodes where they can induce systemic immune responses. Additionally, 4-1BBL maintains immune responses by increasing cytokine and chemokine release by DCs and T cells, expanding T and NK cell populations, preventing activation-induced cell death of T cells, and stimulating memory T cell differentiation[18]. Moreover, infection of tumor cells with LOAd703 enhances tumor cell immunogenicity by upregulation of death receptors and major histocompatibility complex class I (MHC-I)[19].

LOAd703 combined with chemotherapy has been evaluated in patients with advanced pancreatic cancer[20]. Herein, we present the clinical phase I/II trial LOKON003, which was conducted to evaluate safety, activity, and biomarkers of LOAd703 combined with an anti-PD-L1 antibody (atezolizumab) in ICI-resistant CMM patients.

## Results

### Patient characteristics

Between 2021 and 2023, 26 patients were screened for eligibility, of which 24 were recruited (Supplementary Fig. 1): seven to dose cohort $1 \times 10^{11}$ viral particles (VP) and 17 to dose cohort $5 \times 10^{11}$ VP. The data cut was in July 2024. Patient demographics and disease characteristics are summarized in Table 1. The median age was 64 years. Fifty-four percent were male, and 46% were female. At baseline, 63% of patients were assessed as ECOG-0 and the remaining 38% ECOG-1. Fifty percent had elevated lactate dehydrogenase at screening. PD-1 inhibitory treatment alone, or in different combinations, was given in an adjuvant setting in 38% of patients and/or against systemic disease in 92%. Ten of the 24 patients (42%) had received ≥2 prior treatments with a PD-1 targeting antibody (range for all patients: 1–4), including both adjuvant and systemic regimes. The patients had received a median of two prior treatments for systemic disease (range 0-6). Seven patients had a BRAF-mutated tumor, of which five had received prior treatment with BRAF inhibitors before enrollment in this study. Other treatments for systemic disease included experimental ICI combinations, oncolytic virus (T-VEC), chemotherapy and other cell cytotoxic agents as shown in Table 1.

### Safety

Patients received a median of six LOAd703 injections (Table 2), and the most common injection sites were lymph nodes or subcutis/muscles of a limb. The patients received a median of 6 treatments with atezolizumab. Of note, patients in dose cohort $1 \times 10^{11}$ VP received a median of 12 LOAd703 doses and 14 doses of atezolizumab, while the $5 \times 10^{11}$ dose cohort received fewer LOAd703 injections (median 5), as well as atezolizumab infusions (median 5).

The primary endpoint was to evaluate tolerability in terms of adverse events (AEs). AEs attributed to LOAd703 are listed in Table 3. The most frequent AEs were pyrexia, which occurred in 14 patients (58%), chills and vomiting in five patients (21%), and four patients (17%)

had events of cytokine release syndrome (CRS), headache, infusion-related reaction and/or nausea. A vitiligo-like rash was observed in one patient (4%). Overall, most LOAd703-attributed adverse events were mild and transient. There were only two grade 3 dose-limiting toxicities (DLTs) attributed to LOAd703: one case of pyrexia and one injection site reaction, which resolved to baseline. Despite both patients being treated at the $5 \times 10^{11}$ dose cohort, MTD could not be reached as the DLT rate was substantially lower than the target (0.3). No LOAd703 dose reductions were necessary, and no patient experienced LOAd703-attributed AEs resulting in treatment discontinuation. All treatment-emergent adverse events, regardless of an attributed relationship to the study drugs or not, are listed in Supplementary Table 1. There were 17 AEs reported as serious (SAEs,) of which two were attributed to LOAd703 (Supplementary Table 2). These were CRS and pyrexia, both listed as grade 1–2 events.

### Biomarkers and mechanisms of action

Secondary endpoints included the evaluation of biomarkers for mechanisms of action and tumor response. All analyzed patients ($n = 17$) had pre-existing anti-adenovirus antibodies (ADA) which were significantly increased in blood already at week nine post-treatment initiation and thereafter remained stable during study participation (Fig. 1A). There was no significant correlation of the week nine antibody level with the last known date of survival (Fig. 1B). Only 7 of the 22 patients (32%) had detectable levels of antibodies against atezolizumab at any time point post treatment initiation (Fig. 1C). Presence of virus DNA in blood and shedding via urine, saliva, and stool is being evaluated in a cross trial analysis and will be presented elsewhere.

Biopsies from the injected metastases were collected at baseline ($n = 16$) and at week nine post-treatment initiation ($n = 15$). In Fig. 2A, an mRNA heatmap of all patients pre- and post-treatment initiation indicated an induction of immune biomarkers by LOAd703 and atezolizumab. Figure 2B and Supplementary Data 1: NanoString displayed genes that were significantly altered by the treatment, revealing the presence of activated T and NK cells, chemokines attracting them to the TME, as well as receptors that allow them to transmigrate from blood vessels into the TME. Further, markers for antigen presentation and B-cell activation were upregulated. Another observation was that checkpoints such as CTLA-4, TIGIT, LAG-3 and PD-L2 (PDCD1LG2) were amplified as well. Patients were divided according to above or below median OS at baseline (Fig. 2C, E and Supplementary Table 3) and at nine weeks post-treatment initiation (Fig. 2D, Fig. 2F and Supplementary Table 4). When adjusting for multiple testing, no difference in gene expression between the patients below and above the median OS was identified, but the sample size was small when the cohort was divided. As these tests are not compensating for biological relevance (i.e., a small difference may yield a biological difference of one biomarker while another biomarker will need a substantial change in expression for a biological response to occur), there may be false negative samples masked in small data sets. Hence, we analyzed the data set without compensating for multiple comparisons to understand if the data can indicate whether certain biomarkers should be further evaluated in upcoming studies. Without adjusting for multiple comparisons in these smaller sub-cohorts, patients who survived longer than the median had expression of genes involved in antigen presentation and protection against a viral response at baseline, and expression of genes involved in antigen presentation, activation of T and NK cells at week nine post treatment initiation. Instead, patients who survived shorter than the median expressed genes associated with immunosuppressive cells. In blood, protein biomarkers were analyzed with an immune oncology (Fig. 2G and Supplementary Table 5) and an oncology panel (Fig. 2H, Supplementary Table 6). When adjusting for multiple comparisons by the false discovery rate (5%) of Benjamini-Hochberg, only PD-L1 remained significantly

**Table 1 | Patient demographics**

| Characteristics | LOAd703 1 × 10¹¹ VP + Atezolizumab (n = 7) | LOAd703 5 × 10¹¹ VP + Atezolizumab (n = 17) | LOAd703 All + Atezolizumab (n = 24) |
|---|---|---|---|
| Age (median, range, (IQR)) | 70, 50–75 (53–74) | 63, 39–76 (44–71) | 64, 39–76 (51–72) |
| Sex (male/female, %) | 5/2 (71/29) | 8/9 (47/53) | 13/11 (54/46) |
| Melanoma stage IV (%) | 7 (100) | 17 (100) | 24 (100) |
| M1a | 5 (71) | 7 (41) | 12 (50) |
| M1b | 1 (14) | 1 (6) | 2 (8) |
| M1c | 1 (14) | 9 (53) | 10 (42) |
| **Melanoma subtype (*n*, %)** | | | |
| Cutaneous | 7 (100) | 16 (94) | 23 (96) |
| Mucosal | 0 (0) | 1 (6) | 1 (4) |
| Mutations (*n*, %) | 5 (71) | 11 (65) | 16 (67) |
| *NRAS* | 1 (14) | 5 (29) | 6 (25) |
| *BRAF* | 4 (57) | 3 (18) | 7 (29) |
| *KAT6A* | 0 (0) | 1 (6) | 1 (4) |
| **ECOG at baseline (*n*, %)** | | | |
| 0 | 5 (71) | 10 (59) | 15 (63) |
| 1 | 2 (29) | 7 (41) | 9 (38) |
| Elevated LDH at baseline (*n*, %) | 3 (43) | 9 (53) | 12 (50) |
| Prior radiation therapy (*n*, %) | 3 (43) | 8 (47) | 11 (46) |
| Prior adjuvant therapy (*n*, %) | 3 (43) | 6 (35) | 9 (38) |
| Anti-PD-1 only | 3 (43) | 5 (29) | 8 (33) |
| Anti-CTLA-4 then anti-PD-1 | 0 (0) | 1 (6) | 1 (4) |
| Number of prior treatments (for stage IV disease, median, range, (IQR)) | 3, 0–4 (1–3) | 2, 0–6 (1–3) | 2, 0–6 (1–3) |
| Prior treatments for stage IV[a] *n* (%) | 6 (86) | 16 (94) | 22 (92) |
| Anti-PD-1 monotherapy | 3 (43) | 9 (53) | 12 (50) |
| Anti-CTLA-4 monotherapy | 1 (14) | 1 (6) | 2 (8) |
| Anti-CTLA-4 + anti-PD-1 | 2 (29) | 9 (53) | 11 (46) |
| Anti-PD-1 + anti-LAG-3 (experimental) | 0 (0) | 2 (12) | 2 (8) |
| Anti-PD-1 + anti-LAG-3 + anti-Tim-3 (experimental) | 0 (0) | 1 (6) | 1 (4) |
| T-VEC[b] (experimental) | 0 (0) | 1 (6) | 1 (4) |
| T-VEC[b] + PD-1 (experimental) | 1 (14) | 1 (6) | 2 (8) |
| Tyrosine kinase (incl BRAF) inhibitors | 3 (43) | 2 (12) | 5 (21) |
| Temozolomide | 3 (43) | 2 (12) | 5 (21) |
| Paclitaxel + carboplatin | 0 (0) | 1 (6) | 1 (4) |
| Temozolomide + carboplatin | 0 (0) | 1 (6) | 1 (4) |
| Melphalan | 1 (14) | 1 (6) | 2 (8) |
| SGN-CD228A[c] (experimental) | 1 (14) | 0 (0) | 1 (4) |
| LXH254[d] + Ribociclib[e] (experimental) | 0 (0) | 1 (6) | 1 (4) |
| Number of prior anti-PD-1 targeting treatments (median, min–max) | 1 (1–2) | 1 (1–4) | 1 (1–4) |
| Patients receiving ≥2 prior treatments with a PD-1 targeting antibody (*n*, %) | 2 (29) | 8 (47) | 10 (42) |

Source data are provided as a Source Data file.

*VP* viral particles, *IQR* interquartile range, *ECOG* Eastern cooperative oncology group, *LDH* lactate dehydrogenase, increased from reference normal range at respective laboratory; Italic text indicates gene names.

[a]Number of patients receiving ≥1 prior systemic treatment for stage IV disease (range 1–6). prior study participation.
[b]Oncolytic virus.
[c]Antibody-drug conjugate.
[d]pan-RAF kinase inhibitor.
[e]cdk4/6 inhibitor.

elevated ($p = 4.4 \times 10^{-14}$), likely due to the previously reported high PD-L1 in blood after treatment with anti-PD-L1. However, when disregarding multiple comparison compensation for the same reasons as discussed above, biomarkers for the presence and activation of DCs, T- and NK cells were present in blood along with ICI targets such as PD-L1 and PD-1.

## Efficacy

The overall response rate (ORR) was 17% (*n* = 24) (Table 2), with 14% and 18% in dose cohort 1 × 10¹¹ VP and 5 × 10¹¹ VP, respectively. However, 38% (*n* = 9) experienced stable disease for at least eight weeks, resulting in a disease control rate (DCR) of 54% as calculated post-hoc. Of note, in dose cohort 1 × 10¹¹ VP, 57% of the patients showed long-

**Table 2 | Response evaluation**

| Evaluation | LOAd703 1×10¹¹ VP + Atezolizu-mab (n = 7) | LOAd703 5×10¹¹ VP + Atezolizu-mab (n = 17) | LOAd703 All + Atezolizu-mab (n = 24)[b] |
|---|---|---|---|
| Number of LOAd703 injections (median, IQR) | 12 (3–12) | 5 (3–9) | 6 (3–12) |
| **Injection site[a] (n):** | | | |
| Lymph node | 5 | 5 | 10 |
| Liver | 0 | 3 | 3 |
| Abdomen | 0 | 3 | 3 |
| Anterior chest | 0 | 2 | 2 |
| Axilla | 1 | 0 | 1 |
| Leg (gluteus, thigh, knee, lower) | 1 | 7 | 8 |
| Number of atezolizumab infusions (median, IQR) | 14 (3–19) | 5 (3–9) | 6 (3–13) |
| ORR (CR + PR; n, %) (CI) | 1 (14) (CI: 0.007–0.513) | 3 (18) (CI: 0.062–0.410) | 4 (17) CI: 0.067–0.359) |
| Stable disease (SD; n, %) (CI) | 3 (43) (CI: 0.158–0.750) | 6 (35) (CI: 0.173–0.587) | 9 (38) (CI: 0.212–0.573) |
| Disease control (CR + PR + SD; n, %) (CI) | 4 (57) (CI: 0.250–0.842) | 9 (53) (CI: 0.310–0.738) | 13 (54) (CI: 0.351–0.721) |
| Disease control ≥6 months (CR + PR + SD; n, %) (CI) | 4 (57) (CI: 0.250–0.842) | 5 (29) (CI: 0.133–0.531) | 9 (38) (CI: 0.212–0.573) |
| Median PFS (months) (CI) | 9.7 (CI: 2.070–13.90) | 3.0 (CI: 2.140–6.700) | 3.4 (CI: 2.170–7.620) |
| 6-month PFS (n, %) (CI) | 4 (57) (CI: 0.250–0.842) | 5 (29) (CI: 0.133–0.531) | 9 (38) (CI: 0.212–0.573) |
| 12-month PFS (n, %) (CI) | 3 (43) (CI: 0.158–0.750) | 2 (12) (CI: 0.021–0.343) | 5 (21) (CI: 0.092–0.405) |
| Median overall survival (months) (CI) | 26.3 (CI: 2.170–∞) | 12.4 (CI: 4.070–∞) | 19.3 (CI: 4.900–∞) |
| 12-month survival (n, %) (CI) | 4 (57) (CI: 0.250–0.842) | 10 (59) (CI: 0.360–0.784) | 14 (58) (CI: 0.388–0.755) |
| 24-month survival (n, %) (CI) | 4 (57) (CI: 0.250–0.842) | 7 (41) (CI: 0.216–0.640) | 11 (46) (CI: 0.279–0.649) |

Source data are provided as a Source Data file.

*VP* viral particles, *IQR* interquartile range, *ORR* objective response rate, *CR* complete response, *PR* partial response, *SD* stable disease, *PFS*: progression-free survival, *CI* 95% confidence interval, ∞ infinity.

[a]One patient can have more than one injection site.

[b]Patients receiving at least one dose of LOAd703. Data cut-off July 2024.

term disease stability (DCR ≥ 6 months) versus 29% in cohort $5 \times 10^{11}$ VP. Tumor control over time was noted in target lesions (Fig. 3A), including in the LOAd703 injected lesions (Fig. 3B). Nine patients were evaluated with FDG-PET/CT scans, allowing for a post-hoc evaluation of the metabolic ORR. The metabolic ORR was 75% versus 60% in dose cohort $1 \times 10^{11}$ (n = 4) and $5 \times 10^{11}$ (n = 5) VP, respectively (Supplementary Table 7). Progression-free survival (PFS) and overall survival (OS) were also evaluated post-hoc. PFS was 3.4 months (n = 24). Divided by dose cohort, the median PFS was 9.7 and 3.0 months in the $1 \times 10^{11}$ VP and $5 \times 10^{11}$ VP dose cohorts, respectively (Table 2, Fig. 3C). Median OS was 19.3 months (n = 24). Grouped by dose cohort in a post-hoc analysis, the median OS was 26.3 and 12.4 months in the $1 \times 10^{11}$ VP and $5 \times 10^{11}$ VP dose cohorts, respectively (Table 2, Fig. 3D). Of note, the 24-month survival was 57% in dose cohort $1 \times 10^{11}$ and 41% in cohort $5 \times 10^{11}$ VP. In a post-hoc sub-analysis, patients were divided into those who had received one prior treatment (adjuvant or for systemic disease, n = 8) prior to enrollment versus those who received >1 prior treatment (n = 16). Median PFS was 3.0 versus 3.5 months in these groups (Fig. 3E), while median OS was 30.7 versus 10.9 months (Fig. 3F). Patients who received at least one of their LOAd703 injections in a lymph node (n = 10) had poorer median OS compared to patients with other injection sites (10.0 months versus 35.0 months, respectively; Fig. 3G). Patients with M1a status had longer median OS (17.9 months) than M1c patients (12.4 months) as shown in an additional post-hoc analysis, but the curves did not separate (Fig. 3H). There were only two patients with M1b status, and the median OS was not reached. Note that in these post-hoc analyses, the groups were too small to show a statistically significant difference at a 95% confidence interval. In univariate Cox regression analysis, no clinical parameters were associated with OS except for ECOG status at enrollment that was near statistical significance (p = 0.075) as well as receiving four or five previous treatments (p = 0.065 and p = 0.083, respectively; Supplementary Table 8).

### Patients with a particular benefit from treatment

Although there were no complete responses (CRs) during the study treatment period, we observed two patients with partial response (PR) who became tumor-free after study participation. Firstly, a patient with metastases in the axilla, lung, lung hilum, mediastinum, pleura, liver, lymph nodes and subcutis was enrolled. At study week 5, the patient received a thermal ablation of a single pulmonary lesion. After 9 weeks of treatment with LOAd703 and atezolizumab, PR was observed in the non-ablated target lesions, with further response during the study period and completion of all study-related therapy (12 doses of LOAd703 and 19 treatments of atezolizumab). At the end of the treatment period, a complete metabolic response was observed which is maintained currently, without any further treatment, three years after treatment initiation (Supplementary Fig. 2A). Secondly, a patient with metastases in the pectoral muscle and in the subcutis of the abdominal and dorsal side of the trunk experienced a PR at week 18, which eventually deepened throughout therapy. At 11 months post study initiation, a single pectoral metastasis remained, and it was surgically excised (Supplementary Fig. 2B). The patient still remains tumor-free two years and two months after treatment initiation.

## Discussion

In this Phase I/II trial, we investigated LOAd703 in combination with atezolizumab in 24 patients with ICI-refractory stage IV cancer, mainly CMM. The treatment was generally well-tolerated and induced a local and systemic immune response. Both in blood and tumor biopsies, upregulated gene expression of biomarkers related to an ongoing adaptive T cell response. Further, both P-selectin and VCAM1, which

**Table 3 | Adverse events attributed to LOAd703 (n = 24)[a]**

| Preferred term | Total, n (%) of Subjects with AE[b] | | | | | | | | | Total Number of AEs | | |
|---|---|---|---|---|---|---|---|---|---|---|---|---|
| | Grade 1–2 | | | Grade 3 | | | Any grade | | | | | |
| | $1\times10^{11}$ (n = 7) | $5\times10^{11}$ (n = 17) | All (n = 24) | $1\times10^{11}$ (n = 7) | $5\times10^{11}$ (n = 17) | All (n = 24) | $1\times10^{11}$ (n = 7) | $5\times10^{11}$ (n = 17) | All (n = 24) | $1\times10^{11}$ (n = 7) | $5\times10^{11}$ (n = 17) | All (n = 24) |
| Pyrexia | 4 (57) | 10 (59) | 14 (58) | 0 (0) | 1 (6) | 1 (4) | 4 (57) | 10 (59) | 14 (58) | 5 | 16 | 21 |
| Chills | 2 (29) | 3 (18) | 5 (21) | 0 (0) | 0 (0) | 0 (0) | 2 (29) | 3 (18) | 5 (21) | 3 | 4 | 7 |
| Vomiting | 2 (29) | 3 (18) | 5 (21) | 0 (0) | 0 (0) | 0 (0) | 2 (29) | 3 (18) | 5 (21) | 2 | 3 | 5 |
| Cytokine release syndrome | 1 (14) | 3 (18) | 4 (17) | 0 (0) | 0 (0) | 0 (0) | 1 (14) | 3 (18) | 4 (17) | 1 | 3 | 4 |
| Headache | 1 (14) | 3 (18) | 4 (17) | 0 (0) | 0 (0) | 0 (0) | 1 (14) | 3 (18) | 4 (17) | 1 | 6 | 7 |
| Infusion-related reaction | 0 (0) | 4 (24) | 4 (17) | 0 (0) | 0 (0) | 0 (0) | 0 (0) | 4 (24) | 4 (17) | 0 | 7 | 7 |
| Nausea | 3 (43) | 1 (6) | 4 (17) | 0 (0) | 0 (0) | 0 (0) | 3 (43) | 1 (6) | 4 (17) | 6 | 1 | 7 |
| Diarrhea | 0 (0) | 3 (18) | 3 (13) | 0 (0) | 0 (0) | 0 (0) | 0 (0) | 3 (18) | 3 (13) | 0 | 3 | 3 |
| Myalgia | 0 (0) | 3 (18) | 3 (13) | 0 (0) | 0 (0) | 0 (0) | 0 (0) | 3 (18) | 3 (13) | 0 | 3 | 3 |
| Fatigue | 1 (14) | 1 (6) | 2 (8) | 0 (0) | 0 (0) | 0 (0) | 1 (14) | 1 (6) | 2 (8) | 1 | 1 | 2 |
| Pain in extremity | 1 (14) | 1 (6) | 2 (8) | 0 (0) | 0 (0) | 0 (0) | 1 (14) | 1 (6) | 2 (8) | 1 | 1 | 2 |
| Adrenal insufficiency | 0 (0) | 1 (6) | 1 (4) | 0 (0) | 0 (0) | 0 (0) | 0 (0) | 1 (6) | 1 (4) | 0 | 1 | 1 |
| Anemia | 0 (0) | 1 (6) | 1 (4) | 0 (0) | 0 (0) | 0 (0) | 0 (0) | 1 (6) | 1 (4) | 0 | 1 | 1 |
| Back pain | 1 (14) | 0 (0) | 1 (4) | 0 (0) | 0 (0) | 0 (0) | 1 (14) | 0 (0) | 1 (4) | 1 | 0 | 1 |
| Blood creatinine increased | 1 (14) | 0 (0) | 1 (4) | 0 (0) | 0 (0) | 0 (0) | 1 (14) | 0 (0) | 1 (4) | 1 | 0 | 1 |
| Cancer pain | 1 (14) | 0 (0) | 1 (4) | 0 (0) | 0 (0) | 0 (0) | 1 (14) | 0 (0) | 1 (4) | 1 | 0 | 1 |
| Contusion | 0 (0) | 1 (6) | 1 (4) | 0 (0) | 0 (0) | 0 (0) | 0 (0) | 1 (6) | 1 (4) | 0 | 1 | 1 |
| Decreased appetite | 0 (0) | 1 (6) | 1 (4) | 0 (0) | 0 (0) | 0 (0) | 0 (0) | 1 (6) | 1 (4) | 0 | 1 | 1 |
| Dizziness | 0 (0) | 1 (6) | 1 (4) | 0 (0) | 0 (0) | 0 (0) | 0 (0) | 1 (6) | 1 (4) | 0 | 1 | 1 |
| Dysgeusia | 0 (0) | 1 (6) | 1 (4) | 0 (0) | 0 (0) | 0 (0) | 0 (0) | 1 (6) | 1 (4) | 0 | 1 | 1 |
| Embolism | 0 (0) | 1 (6) | 1 (4) | 0 (0) | 0 (0) | 0 (0) | 0 (0) | 1 (6) | 1 (4) | 0 | 1 | 1 |
| Groin pain | 0 (0) | 1 (6) | 1 (4) | 0 (0) | 0 (0) | 0 (0) | 0 (0) | 1 (6) | 1 (4) | 0 | 1 | 1 |
| Hematuria | 1 (14) | 0 (0) | 1 (4) | 0 (0) | 0 (0) | 0 (0) | 1 (14) | 0 (0) | 1 (4) | 1 | 0 | 1 |
| Hypotension | 1 (14) | 0 (0) | 1 (4) | 0 (0) | 0 (0) | 0 (0) | 1 (14) | 0 (0) | 1 (4) | 1 | 0 | 1 |
| Hypoxia | 0 (0) | 1 (6) | 1 (4) | 0 (0) | 0 (0) | 0 (0) | 0 (0) | 1 (6) | 1 (4) | 0 | 1 | 1 |
| Injection site pain | 0 (0) | 1 (6) | 1 (4) | 0 (0) | 0 (0) | 0 (0) | 0 (0) | 1 (6) | 1 (4) | 0 | 1 | 1 |
| Injection site reaction | 0 (0) | 0 (0) | 0 (0) | 0 (0) | 1 (6) | 1 (4) | 0 (0) | 1 (6) | 1 (4) | 0 | 1 | 1 |
| Lymphocyte count decreased | 0 (0) | 1 (6) | 1 (4) | 0 (0) | 0 (0) | 0 (0) | 0 (0) | 1 (6) | 1 (4) | 0 | 1 | 1 |
| Nasopharyngitis | 0 (0) | 1 (6) | 1 (4) | 0 (0) | 0 (0) | 0 (0) | 0 (0) | 1 (6) | 1 (4) | 0 | 1 | 1 |
| Neuropathy peripheral | 0 (0) | 1 (6) | 1 (4) | 0 (0) | 0 (0) | 0 (0) | 0 (0) | 1 (6) | 1 (4) | 0 | 1 | 1 |
| Edema peripheral | 1 (14) | 0 (0) | 1 (4) | 0 (0) | 0 (0) | 0 (0) | 1 (14) | 0 (0) | 1 (4) | 1 | 0 | 1 |
| Paresthesia | 0 (0) | 1 (6) | 1 (4) | 0 (0) | 0 (0) | 0 (0) | 0 (0) | 1 (6) | 1 (4) | 0 | 1 | 1 |
| Tachycardia | 1 (14) | 0 (0) | 1 (4) | 0 (0) | 0 (0) | 0 (0) | 1 (14) | 0 (0) | 1 (4) | 2 | 0 | 2 |
| Troponin I increased | 0 (0) | 1 (6) | 1 (4) | 0 (0) | 0 (0) | 0 (0) | 0 (0) | 1 (6) | 1 (4) | 0 | 1 | 1 |

Data are n (%) unless otherwise specified. The worst grade is included in the table. There were no grade 4 or 5 events.

*AE* adverse event.

[a]Safety-evaluable patients.

[b]Related AEs. Data cut-off December 2023.

support their transmigration into the TME, were upregulated. Additionally, markers for antigen presentation via DCs and B cells were upregulated. Notably, ICI targets were simultaneously upregulated in injected metastases, which indicates a benefit from combining LOAd703 with additional ICIs other than atezolizumab. Patients who survived longer also had high levels of genes related to adaptive immunity as well as genes that associate with protection against a viral response, such as CD55, toll-like receptor 6 (TLR6) and interferon

alpha 7 (IFNa7). Hence, not only the transgenes but the adenoviral backbone may be of importance to drive anti-tumor activity post LOAd703 treatment. In contrast, in the population with shorter than median survival, genes associated with T regulatory cell recruitment and T cell tolerance, angiogenesis, myeloid transformation, cancer-associated fibroblasts, and activation of latent TGFbeta such as uro-kinase receptors (uPAR, PLAUR) were expressed post treatment initiation. However, these small sub-cohorts were analyzed without

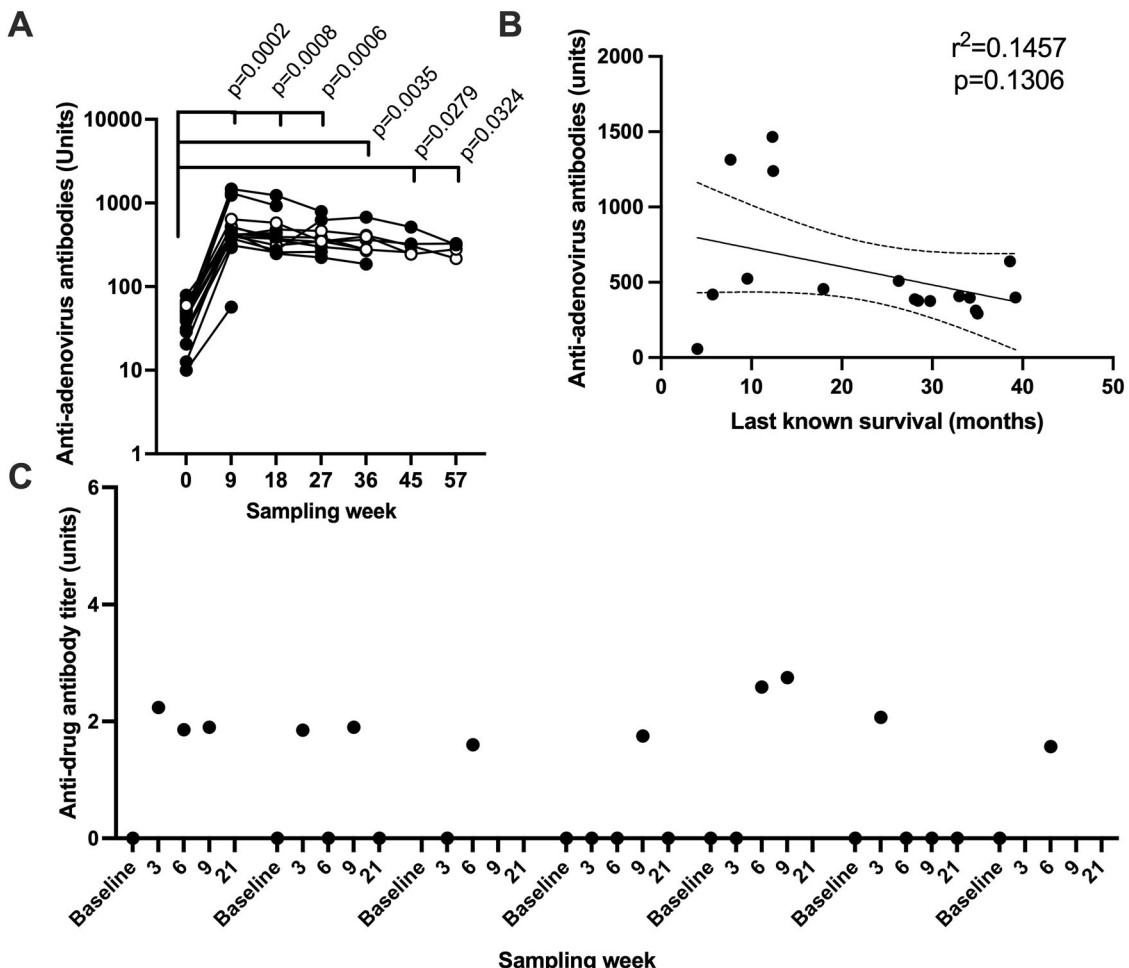

**Fig. 1 | Anti-drug antibodies.** Source data are provided as a Source Data file. **A** Anti-adenoviral antibodies were evaluated in blood over time and evaluated by ELISA ($n = 17$ at baseline). Filled circles represent dose level $5 \times 10^{11}$VP (viral particles) and open circles represent $1 \times 10^{11}$VP. Significant changes were evaluated by mixed-effects analysis for repeated analyses with multiple comparisons, alpha 0.05. **B** Antibody units at week nine were correlated to the last known survival date (ten patients censored from 17 months and onwards) using a Simple linear regression analysis at a significance level of 0.05. **C** Anti-drug antibodies to atezolizumab were detected in seven of 22 patients receiving at least one dose of atezolizumab. The figure displays anti-drug antibodies in those seven patients at baseline and at multiple time points post-treatment initiation.

multi-comparison compensation, and results need to be validated in larger cohorts where multi-comparison adjustment can be applied or with singleplex analytical methods. Of note, soluble PD-L1 was significantly detected in high quantities post-treatment initiation, also using multi-comparison compensation. It remains unclear whether the PD-L1 level is increased due to PD-L1 upregulation, increased shedding from cells, or reduced soluble PD-L1 clearance from the body due to complex formation between with atezolizumab and PD-L1. In a larger study also including a control group, it would be of interest to understand if there are differentially expressed biomarkers at baseline that can predict patients who will respond and thereby support patient stratification post-marketing. Considering the immunosuppressive phenotype of patients with shorter survival than median, a combination treatment with agents that reduce immunosuppressive myeloid cell populations and T regulatory cells is of interest. However, the increase of LAG-3 post-treatment initiation indicates that adding the approved combination of PD-1 and LAG-3 ICIs to LOAd703 may be a suitable next step in malignant melanoma and may also open the way for an earlier line of treatment.

The survival data in the study proved to be particularly noteworthy as the lower dose cohort experienced a median OS longer than two years, and the higher dose cohort over one year. However, a majority of patients (five out of seven; 71%) in the lower dose cohort

were staged as M1a, while seven out of 17 (41%) were M1a in the higher dose cohort, which may have impacted the survival difference. Nevertheless, compared to real-world data, OS seemed superior compared to chemotherapy or best supportive care, which would have been the alternative options for the vast majority of the study participants, with an OS of six months is expected[21]. In a Phase II study by Chesney et al., patients with ICI-refractory CMM treated with TILs (lifileucel, Amtagvi®) experienced a median OS of 13.9 months[22]. In a parallel phase III study by Rohaan et al., TIL therapy was compared to an ipilimumab control arm. The TIL arm showed a median OS of 25.8 months[23]. This patient population had better prognostic factors compared to the present study, with only 20% of patients having elevated LDH (compared to 50% in this study), and 11% of the patients had not received previous systemic therapy. TIL therapy requires tumor resection, lymphodepletion with chemotherapy, and the addition of IL-2, all with associated morbidity and toxicity. In the study by Chesney et al., grade 3-4 TEAEs occurred in over 30% of patients, including febrile neutropenia in 42%[22]. In comparison, LOAd703 had an advantageous toxicity profile and could be a suitable choice for patients unfit for surgery and chemotherapy. Nevertheless, the sample size in the current study was small, and due to the limitations of the study design (open-label without a control group), efficacy results should be interpreted with caution.

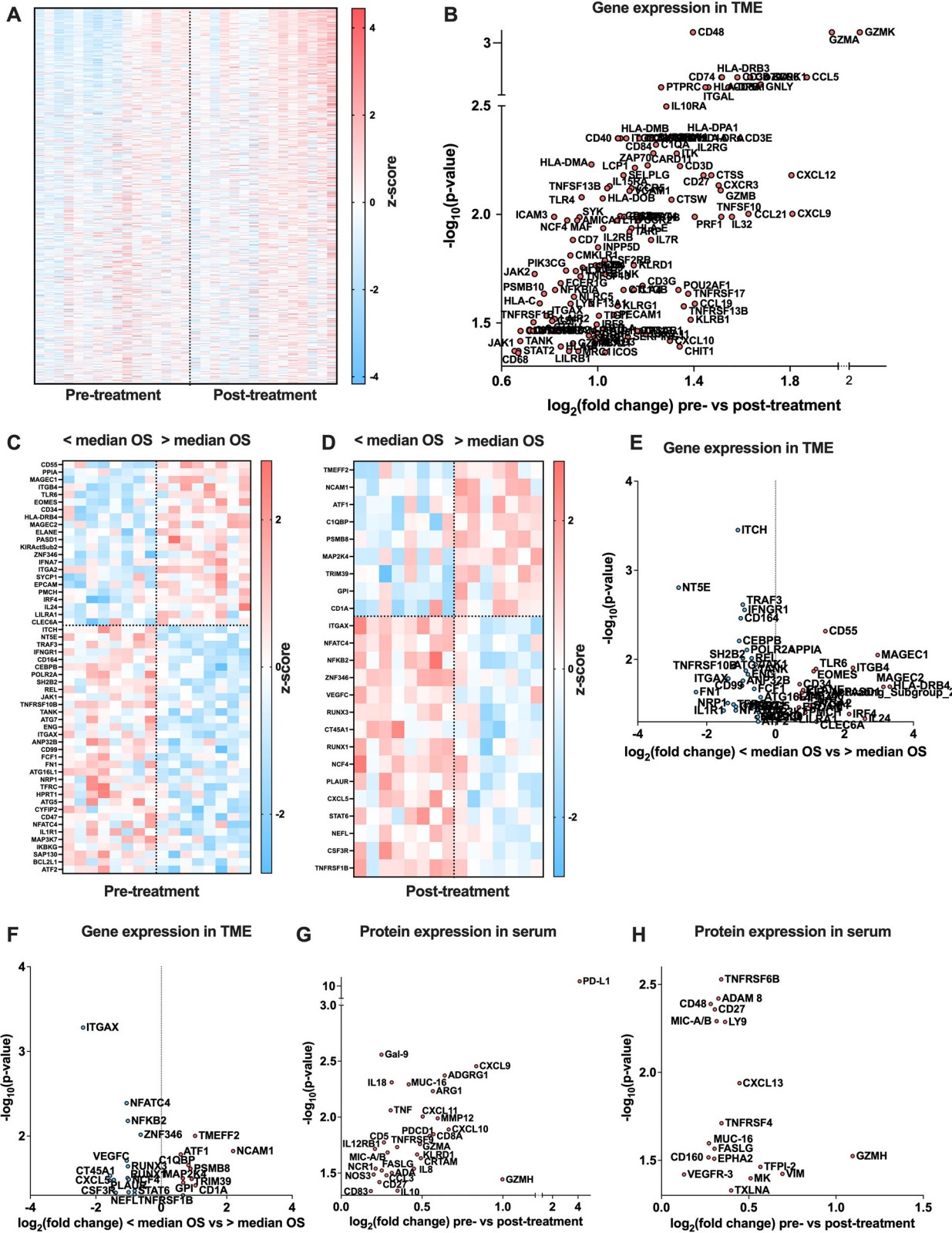

Since LOAd703 is administered intratumorally, the negative impact of the increasing anti-LOAd703 antibodies (ADA) is minimized. Indeed, anti-LOAd703 antibody titers did not significantly correlate with the last known date of survival. However, ten patients were censored but as they were all censored past 17 months, a longer survival would not change the outcome of this analysis as their ADA levels at week 9 were similar to other patients' independently of survival (below 1000 units), with the exception of three patients with higher ADA titers, above 1000 units, who died within 7-12 months post enrollment. There was no indication that LOAd703 altered the formation of ADA against atezolizumab as the ADA titers increased during the study from 0 % to 32 % (7/22) of patients, falling within the previously reported range of ADA incidences during atezolizumab monotherapy (13-54 %)[24].

**Fig. 2 | Immune markers.** Gene expression in tumor biopsies (tumor microenvironment: TME) at baseline (pre; $n = 16$) and at week nine post treatment initiation (post; $n = 15$) was measured using NanoString. **A** Gene expression shown as a z-score heatmap. **B**) Volcano plot displaying statistically significant alterations in fold change (pre vs post) gene expression. Statistics were done with RUV-4 with correction for multiple comparisons by the false discovery (5%) rate method of Benjamini-Hochberg (two-sided). **C–F** Patients were dichotomized according to above or below median overall survival (OS), and differences in gene expression levels were compared between the two groups using two-sided unpaired $t$ tests without correction for multiple comparisons based on RUV-4 approximated data.

Genes significantly differently expressed in the two groups are shown as z-score heatmaps (**C–D**) and volcano plots (**E–D**) pre- (**C** and **E**) and post- (**D** and **F**) treatment. In volcano plots, red indicates genes that are highly expressed and blue indicates genes that are less expressed in patients with an above median OS compared to patients with a below median OS. **G, H** Protein detection in serum was evaluated with the Olink Immuno-oncology (**G**, $n = 16$) and Oncology II (**H**, $n = 17$) proteomic panel and results displayed in volcano plots as fold change: week nine post-treatment initiation values divided by baseline values for proteins significantly altered by the treatment. Statistics were performed by multiple paired, two-sided $t$-tests with no correction for multiple testing.

Two viral products have been approved for TME gene engineering. The first, talimogene laherparepvec (Imlygic®), is a replication-competent herpes simplex virus encoding granulocyte macrophage-colony stimulating factor (GM-CSF) approved for melanoma[25]. The second, nadofaragene firadenovec-vncg (Adstiladrin®) is a replication-defective adenovirus encoding interferon alpha 2b (IFNalpha2b) and is approved for non-muscle invasive bladder cancer[26]. Antibodies targeting immune activating pathways via receptors such as CD40, 4-1BB and OX40 have shown interesting results in preclinical and early clinical studies. However, systemic exposure of such antibodies at clinically relevant doses has been limited due to toxicities[9,27,28]. For example, dose escalation of CD40 monoclonal antibodies is limited due to the occurrence of hepatic toxicity[9]. Further, as CD40 is present on many cell types, the antibodies may bind to irrelevant targets instead of selectively targeting DCs in TME[9].

In conclusion, the current study indicates that LOAd703 in combination with ICI is a strong potential candidate for future treatment of advanced CMM patients by reversing the ICI-induced resistance. The treatment was well tolerated, and the anti-tumor effects are promising.

## Methods
### Study design
The study was a multi-center, single-armed, open-label, dose-escalation Phase I/II clinical trial (LOKON003; NCT04123470) evaluating the safety and effect of LOAd703 in combination with atezolizumab in patients with unresectable or metastatic melanoma at three recruiting centers: Uppsala University Hospital (Uppsala, Sweden), The Angeles Clinic and Research Institute, A Cedars Sinai Affiliate (Los Angeles, CA) and Baylor College of Medicine (BCM, Houston, TX). The study followed a Bayesian Optimal Interval (BOIN) design testing LOAd703 at two dose levels ($1 \times 10^{11}$ and $5 \times 10^{11}$ VP) with a target DLT rate of 0.3 and a maximum of 25 patients enrolled at the maximum tolerated tested dose (MTD). The DLT period encompassed at least two doses of LOAd703, with at least 3 weeks of AE evaluation post-second dose. Sample size was determined based on an assumed response rate of 30% (Study Protocol is found in Supplementary Data 1). The study was reviewed and approved by the ethical review boards Etikprövningsmyndigheten (Uppsala University Hospital), WCG IRB (The Angeles Clinic and Research Institute) and the IRB at Baylor College of Medicine (BCM) (Sweden DNr 2019-05974, 2019-12-18; US IRB: 20193343, 2020-01-03). It was conducted in accordance with the ICH Harmonized Tripartite Guideline for Good Clinical Practice and the principles of the Declaration of Helsinki. All patients provided written informed consent and had the right to withdraw consent to further study participation at any time.

### Patient enrollment and eligibility
Enrolled patients had stage IV CMM ($n = 23$) or mucosal melanoma ($n = 1$) with documented disease progression after one or several treatments with a PD-1 inhibitor. Swedish patients with BRAF-mutated melanoma were required to have received previous BRAF and/or MEK inhibitor therapy. Other previous treatments were allowed. Patients were required to have measurable disease according to RECIST 1.1

criteria with at least one visible injectable tumor lesion exceeding 10 mm on CT scan or MRI. If injected tumor lesions had been irradiated, confirmed radiological growth after irradiation was required. Age >18 years, Eastern Cooperative Oncology Group Performance Status (ECOG) performance status 0–1, and acceptable blood laboratory values (hematology, coagulation, liver aminotransferases and serum albumin) were also prerequisites for study participation.

Patients were excluded if they had rapid disease progression at the time of enrollment, clinically active cerebral metastases, a history of leptomeningeal disease, another malignant disease within the past two years, and concurrent or recent medication that could increase toxicity or interfere with the study treatment (including corticosteroids in doses of >10 mg prednisone equivalent per day). Further, patients with active, severe autoimmune disease or a history of grade 4 immune-related adverse events from earlier ICI monotherapy that had required steroid treatment for >12 weeks were also excluded. Patients with uncontrolled intercurrent illness that could compromise compliance with study requirements were ineligible.

### Procedures
Enrolled patients were treated with LOAd703 ($1 \times 10^{11}$ or $5 \times 10^{11}$ VP; manufactured by Baylor College of Medicine, TX) administered by image-guided, percutaneous intratumorally injection every third week. Atezolizumab (provided by Hoffmann-La Roche) was administered intravenously at a fixed dose of 1200 mg every third week. Patients could receive up to 12 cycles of combined treatment followed by up to seven infusions of atezolizumab monotherapy, resulting in a total maximal study treatment duration of 57 weeks.

Adverse events were collected throughout study participation and evaluated accordingly to the National Cancer Institute Common Terminology Criteria for Adverse Events (CTCAE v.5.0). Radiological evaluation was performed at baseline and every eight weeks. Patients with disease burden localized to the chest and abdominal area were evaluated with CT scans only. Patients with disease burden localized to the limbs or head and neck area were evaluated with FDG-PET/CT scans, and response was determined accordingly to RECIST 1.1. while the metabolic response was evaluated according to the European Organization for Research and Treatment of Cancer (EORTC). After the end of study participation, disease progression and survival were monitored every four months until death, withdrawal of consent or study end. Samples for ADA and biomarkers were taken at baseline and at multiple timepoints as indicated in Supplementary Fig. 1.

### Objectives and endpoints
The primary objective of the study was to evaluate the tolerability of LOAd703 in combination with atezolizumab, determined by evaluation of adverse events according to CTCAE v5.0.

The secondary objectives were to determine anti-tumor activity, pharmacokinetics, and biological mechanisms-of-action of LOAd703 in combination with atezolizumab, including biomarker, shedding and anti-drug antibodies (PK and shedding are evaluated in a cross-trial evaluation and will be presented elsewhere). The endpoint for anti-tumor activity is ORR. Further, disease control (CR + PR + stable

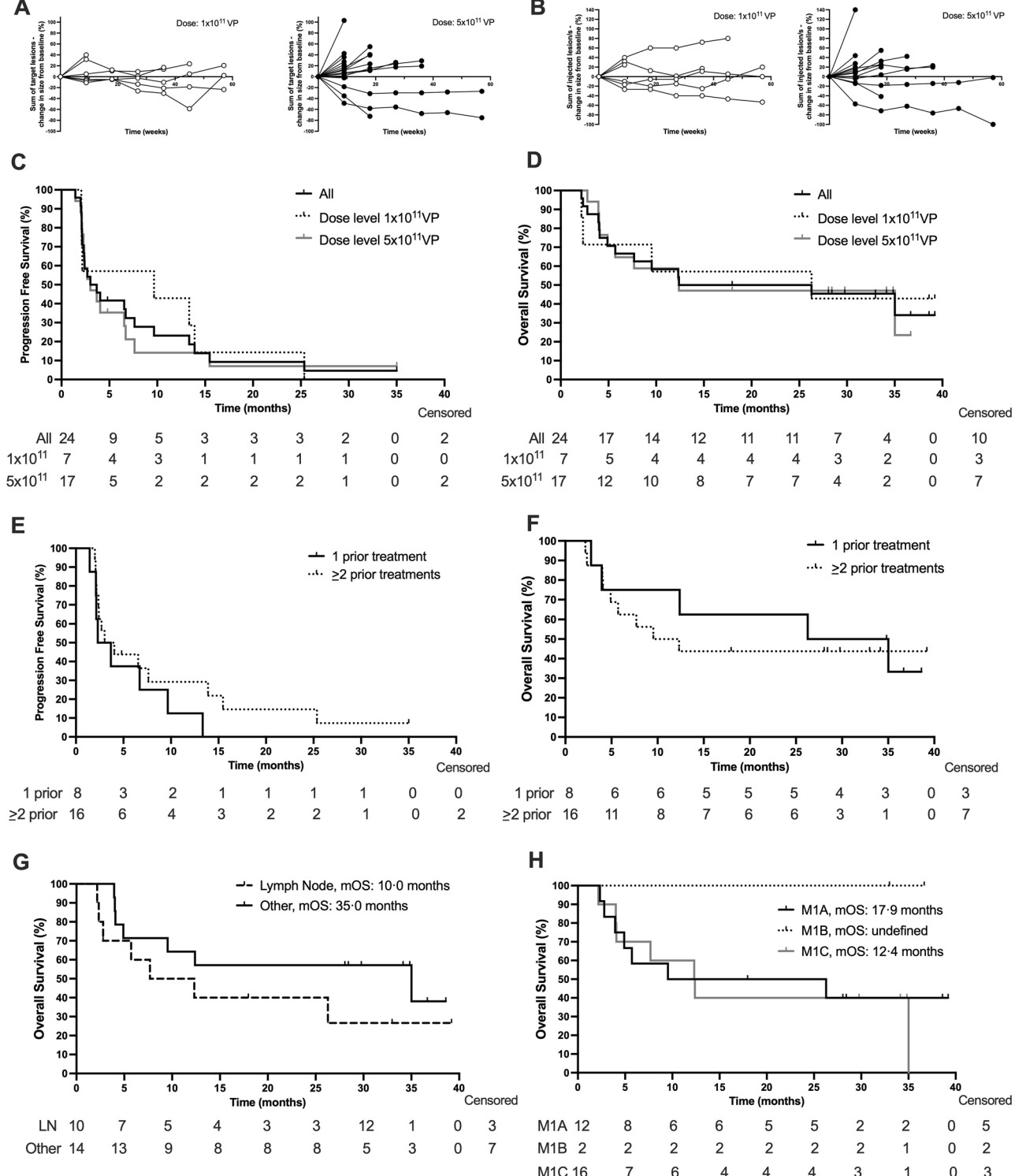

**Fig. 3 | Response evaluation.** Source data are provided as a Source Data file. The percent change in target lesion sizes (**A**) or injected lesion size (**B**) in individual patients from baseline, divided by LOAd703 dose level. **A** Open circles: $1 \times 10^{11}$ viral particles (VP), $n = 6$, and closed circles: $5 \times 10^{11}$ VP, $n = 17$. **B** Open circles: $1 \times 10^{11}$ VP, $n = 6$, and closed circles: $5 \times 10^{11}$ VP, $n = 15$. Patients with at least one imaging post-baseline were included in the analysis. **C** Progression-free survival grouped by LOAd703 dose level. **D**) Overall survival grouped by LOAd703 dose level. **E** Progression-free and **F** overall survival for patients having received one prior line of treatment versus ≥ two previous treatment lines. For **C** to **F** Patients who received at least one dose of LOAd703 were included in the analysis ($1 \times 10^{11}$ VP, $n = 7$; $5 \times 10^{11}$ VP, $n = 17$, all patients, $n = 24$). Kaplan–Meier curves displaying median overall survival (mOS) divided by **G** LOAd703 injection sites (lymph node, $n = 10$; other sites, $n = 14$) or **H** M1 status (M1a, $n = 12$; M1b, $n = 2$; M1c, $n = 10$). Of note, patients may have been injected in more than one lesion. The lymph node group of patients had received at least one injection in a lymph node. Patients at risk are listed below the graph at each time point, along with the number of censored patients at the study endpoint. Data cut-off July 2024.

disease (SD)) and ≥6 months disease control were calculated post-hoc. Additionally, the median PFS and median OS were calculated post-hoc. Date of enrollment to date of progression (including clinical progression at the discretion of the investigator) or date of death was used to calculate PFS, while date of enrollment to date of death was used to calculate OS. Patients still alive or lost to follow-up at the cut-off were censored.

### Anti-drug antibody analyses

Serum from patients collected at baseline and at several timepoints during study participation was analyzed by an anti-human adenovirus (Hxn) IgG ELISA kit (Alpha Diagnostic, San Antonio, TX, USA) accordingly to the company instructions. Samples were analyzed with a 3-tiered approach (screening, confirmation and titer) to detect antibodies against atezolizumab using a service provided by ICON plc (Farmingdale, NY, USA). Significant changes were evaluated by mixed-effects analysis for repeated analyses, alpha 0.05, and antibody units at week nine were correlated to the last known survival date (10 patients censored from 17 months and onwards) using a Simple linear regression analysis at a significance level of 0.05. Both analyses were done using GraphPad Prism 10.4.1 (La Jolla, CA, USA).

### Immune biomarkers

Tumor biopsies were collected at baseline and at week nine post-treatment initiation. From tumor biopsies, mRNA was purified using the RNeasy Mini Kit (QIAGEN, Hilden, Germany). The mRNA from each patient was analyzed with NanoString technology (NanoString Technologies, Seattle, WA, USA) using the nCounter® PanCancer Immune Profiling Panel and the assay was performed by the Clinical Genomics Unit at the Uppsala University Hospital, Uppsala, Sweden. Data was normalized with RUV-4 in RStudio for the fold change between baseline and week 9, and then the approximate individual value was calculated by removing unwanted variation from the observed $\log_2$ value. For evaluation of alterations in gene expression post-treatment, multiple paired $t$ tests were conducted with correction for multiple comparisons by the false discovery rate (5%) of Benjamini-Hochberg. For examination of differences in gene expression pre and post-treatment based on OS, unpaired $t$-tests were conducted. Because of the small sample size of the compared groups, no adjustment for multiple comparisons was applied.

Plasma from patients collected at baseline and at week nine post-treatment initiation was analyzed with the Olink Target 96 Immuno-Oncology panel and the Olink Target 96 Oncology II (OlinkProteomics AB, Uppsala, Sweden). The assay was performed by the Olink Proteomics Service Facility, Uppsala, Sweden. The difference in plasma protein levels pre and post-treatment was assessed by a paired $t$-test. Statistical testing was done using RStudio or GraphPad Prism version 10.4.1.

### Statistical analyses for clinical assessments

Descriptive statistics were presented with median, range and interquartile range (IQR) for continuous variables. Categorical variables were enumerated by numbers and percentages when appropriate. Efficacy prediction (sample size) was evaluated using a one-sided binomial test with a significance level of 0.025. At a maximum enrollment of 25 patients at the maximum tolerated dose (MTD), the null hypothesis could be rejected with a probability of 0.9 given an assumed true response rate of ≥30% (Study protocol: https://clinicaltrials.gov/study/NCT04123470?term=LOKON003&rank=1). PFS and OS were assessed using the Kaplan-Meier estimator for survival analysis. Cox regression analysis was conducted in R Studio IDE, AGPL v3.

### Reporting summary

Further information on research design is available in the Nature Portfolio Reporting Summary linked to this article.

## Data availability

Data sharing requests will be considered on a case-by-case basis to comply with the protection of personal data regulations. Requests by academic study groups for deidentified patient data with the intent to achieve aims of the original proposal can be forwarded to the corresponding author, who will evaluate if the request is in line with approved use by Ethical Review Boards and Regulatory Bodies, and if so, share a data file to be used by the academic study groups under a signed agreement. The Study Protocol is available in Supplementary Information). Source data are provided (Source Data), other data is provided in the Article and/or Supplementary Information. Source data are provided with this paper.

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

## Acknowledgements

The authors acknowledge the research teams at the respective Hospital and University for patient care and sampling patients. The graphical abstract was made using BioRender. The trial was sponsored by Lokon Pharma AB, while the research analyses were funded by grants to Dr Loskog and Dr Ullenhag from the Swedish Cancer Society (23-2698, 22-2342), and grants to Dr Loskog from the Swedish Research Council (2024-03471), and grants to Dr Loskog and Lövgren from the Swedish Childhood Cancer Fund (2021-0061). Lokon Pharma was involved in study design, analysis, and writing the manuscript as stated in the author contributions.

## Author contributions

A.L., D.W., G.U., and O.H. designed the trial and wrote and/or commented on the protocol. J.L.J. and L.S. finalized the SAP and coordinated the study, data management and safety reporting. G.U., V.E.R., O.H., I.M., D.W., M.P., S.A., and S.I. treated the patients, handled adverse events and performed response evaluations. E.E. and T.L. organized the plan for laboratory endpoints and executed the analysis. H.G.W. and C.N. executed the Nanostring and Olink proteomics analyses under supervision by A.L. A.L., G.U., and V.E.R. wrote the first draft of the manuscript, which was revised by all authors. L.S. performed a QC check on all data. The final manuscript was approved by all authors.

## Funding

## Competing interests

Drs. Loskog, Eriksson, Sandin, Leja-Jarblad, and Ms. Norström are employees of Lokon Pharma AB. Dr. Loskog is a board member of Lokon Pharma AB and an alternate board member of Nexttobe AB. Further, she is the inventor of a patent owned by Lokon Pharma AB in regard to LOAd703 and holds a research grant together with Dr. Lövgren from Lokon Pharma AB. Other authors do not report any conflicts of interest in regard to the data presented in the paper.
