## [Transparent Peer Review file · Nature Communications]

LOAd703-Induced Tumor Microenvironment Gene Engineering in Combination with Atezolizumab in Metastatic Malignant Melanoma: A Phase I/II Trial

Corresponding Author: Professor Angelica Loskog

Version 0:

Reviewer comments:

Reviewer #1

(Remarks to the Author)

The authors have to some extent addressed my comments including adding information on M1 stages which points towards marked differences in subgroups among the two patient cohorts. Thus, the low dose cohort includes only 14 % M1c patients while the high dose has 41% M1c. The authors have added text and a figure on survival according to M1 stage in the result section. However, the total number of patients as well as the numbers of patients in the different M1 stages do not harmonize between table and figure/figure legend. Table: M1b: n=11, M1b: n=5, M1c n=:8, with a total of 24 patients. Figure 3H: M1b: n=12, M1b: n=2, M1c: n=16 with a total of 30 patients. Figure 3H legend: M1c, n=12; M1c, n=2; M1c, n=10 with a total of 24 patients. If the numbers given in the table is to be trusted then new calculations of survival in Figure 3H needs to be carried out and associated text revised.

Further, the authors have now included a cox regression analysis claiming that “ no clinical parameters impacted on survival”. However, from the related suppl tabl S9 it appears that M1 stage is not included. M1 staging is one of the strongest prognostic indicators for survival and should be included in the cox analysis.

The impact of M1 stage in regard to interpretation of results between VP dose cohorts in the trial, and in comparison with other trials, have not been integrated in the Discussion. It is recommended to incorporate a paragraph on trial limitations highlighting this together with the potential impact of post PD treatment on OS as previously mentioned.

Reviewer #2

(Remarks to the Author)

I thank the authors for their comprehensive replies.

In the discussion, I still miss some future perspectives with this treatment strategy. Should this be investigated in earlier treatment lines or in combination with other drugs? Could this be a biomarker-driven drug and if so which one? The authors should scientifically comment on this regardless on strategic decisions of the biotech company producing the drug.

Furthermore, I have some small spelling suggestions:

INTRODUCTION:

Immune checkpoint inhibition (ICI) targeting PD-1, CTLA-4, and LAG-3, HAS revolutionized ... and IS standard-of-care together with protein kinase inhibitors for BRAFV600E-mutated CMM

DISCUSSION:

additional ICI than atezolizumab

Reviewer #3

(Remarks to the Author)

The paper reports results from a Phase I/II trial that employed a BOIN design for dose escalation across two dose levels. The trial planned to enroll 25 evaluable subjects at the MTD (up to 50 total), but was terminated early due to lack of efficacy signal. Dose escalation occurred after seven patients were enrolled at the initial dose level. Given the small sample size ($N = 24$), all analyses are exploratory and should be interpreted with caution, as noted by the authors. Nevertheless, the statistical reporting requires greater rigor and improvement. Specific recommendations include:

1. The rationale for $N = 25$ appears to be based on efficacy assumptions. However, the statement lacks key details, such as the assumed true response rate under the alternative hypothesis. The study protocol includes this information; authors should reference the protocol as supplementary material or cite the published protocol if available.
2. All point estimates (e.g., ORR, PR, SD rates, median PFS/OS) should be accompanied by 95% confidence intervals, as specified in the protocol.
3. Statistical methods used for all analyses should be described in the main text. For example, methods mentioned in figure titles (Figures 1–2) should be expanded with sufficient detail.
4. AE table (Table 3) should present adverse events by dose level, consistent with the study SAP.
5. Figure 1A compares anti-adenoviral antibody levels over time by dose level. The use of two-way ANOVA for longitudinal measurements is not appropriate; a repeated-measures or mixed-effects model would be more suitable.
6. Figure 1B correlates antibody units at week nine with overall survival using simple linear regression, which is inappropriate given the presence of censored survival data. A Cox proportional hazards model or similar approach should be considered.
7. It is unclear why some analyses in Figure 2 include multiple comparison adjustments while others do not. Adjustment should be applied consistently across marker comparisons.
8. Information on study drug compliance is missing and should be included.

Minor Comments

1. Phrases such as “significant difference at a 95% confidence interval” are unconventional. Recommend revising to “at a significance level of 0.05.”
2. In the Results section (page 4, lines 85–89), some numbers cited in the text do not appear in Table 1.
3. In the Methods section titled “Statistical analyses for clinical assessment” (page 19, line 522), the term “one-sided binominal t-test” is unclear. For a binary endpoint like ORR, a one-sided binomial test would be appropriate, but not a t-test.

Reviewer #4

(Remarks to the Author)

Version 1:

Reviewer comments:

Reviewer #1

(Remarks to the Author)

I have no further comments

Reviewer #3

(Remarks to the Author)

We thank authors for fully addressing almost all of our prior comments. The only revision needed is about the reported confidence intervals. By convention, the confidence interval is written as (lower boundary, upper boundary) or lower boundary-upper boundary. But some CIs currently have upper boundary reported first followed by lower boundaries. These should be corrected.

Reviewer #4

(Remarks to the Author)

Responses to the previous comments from Nature Medicine's referees

Reviewer #1 (Remarks to the Author):

Comment 1

LOAd703-Induced Tumor Microenvironment Gene Engineering in Combination with Atezolizumab in Metastatic Malignant Melanoma – A Phase I/II Trial. Hamid et al.

The authors report from a phase I/II trial where 24 patients were treated with a CD40L and 4-1BBL gene modified oncolytic virus at two dose levels in combination with atezolizumab. The rationale for this strategy is substantiated and represents an interesting approach. The manuscript, however, suffers from some weaknesses. Thus, important data on prognostic patient characteristics is lacking and further the authors are prone to unsubstantiated interpretation of efficacy results despite the low patient numbers and the insufficient prognostic information.

According to the protocol 'At least 25 response evaluable patients will be enrolled at the MTD for evaluation of their response'. Please explain the reason for closing the trial after 17 patients at the highest dose level? And add information of this deviation in the manuscript including how this affects the validity of the reported ORR according to the statistical considerations in the protocol.

Response: The study was the first trial treating melanoma patients with LOAd703 and also the first study combining LOAd703 with atezolizumab. Hence, the early stage character of the study called for a cautious design with an appropriate number of patients. The statistical design built on overall response rate which in turn considers only CRs and PRs. After enrolling a total of 24 response evaluable patients (7 at dose level 1 and 17 at dose level 2) we realized that the ORR was not as high as we stipulated in the statistics and enrolling one last patient at dose level 2 would not make us reach the primary endpoint independently of response. Hence, the study was stopped. However, we also learnt that the dose level 1 may be better than dose level 2 in terms of survival. The mode-of-action of our treatment called for another type of study design based on tumor control and overall survival as the next step. We believe this was the correct decision for this study and our results show the importance that survival also in earlier studies, perhaps more important for immunotherapy studies that can induce long term tumor control beyond treatment termination due to the immunological memory achieved.

Comment 2

All patients are reported to have stage IV disease, however, it appears from table 1 that for 71% of the patients the M1 stage is not specified. This information is crucial for the interpretation of the results and should be included in the manuscript. Especially as the remaining 29 % of the patients are reported to be the good prognosis stage M1a and b and a potential unbalance of stage M1c between the two dose levels might affect the efficacy measures and considerations.

Response: We agree that the M stages should be mentioned to make comparisons easier and have included that in Table 1.

Comment 3

The authors spend quite some efforts in Results and Discussion interpreting the efficacy data especially PFS and OS including comparison between the two dose levels as well as comparison with efficacy data from published trials on e.g. TIL therapy. The authors should be more cautious in their interpretation especially when claiming comparability in patient characteristics when e.g. a major prognostic parameter like M stage is unspecified in the majority of the trial patients. Further, post-trial therapy might significantly impact OS e.g. BRAF/MEKi and TIL therapy or ipi/nivo

for patients primary treated with either mono anti-PD1 or Opdivo. In order to speculate on the OS observed data on post-trial therapy should be included in the paper and taken into consideration.

Response: We disagree that we discuss the data in the Results section but we do report ORR, PFS and OS, as most trials do even if numbers are small in early studies. Further, the readers are appropriately notified that this is an early study and the number of patients enrolled. As this is an early study, it is not randomized. Therefore, in the discussion we are putting our data into context by describing effect data in similar patient populations. This is also commonly done in the discussion section which is supposed to be a discussion of your data and other researchers findings in the same topic.

Comment 4

Page 3; 'The TME consists of multiple cell types including fibroblasts, vascular endothelium, immune cells, and products released by them such as chemokines, extracellular matrix and vesicles, that collectively induce immunosuppression', recommend adding 'cytokines'.

Response: as it is many types of factors released, not only cytokines, that affect the suppressive nature, we stated "products released by them" instead in sake of keeping the introduction a bit shorter.

Comment 5

Page 4, Patient Characteristics: 'All patients had stage IV CMM', recommend to add ' progressive'

Response: We have shortened the results and the same sentence is not included anymore, however, in the methods the word documented disease progression is mentioned.

Comment 6

Page 5, Safety: 'the most common injection sites were lymph nodes or subcutis/muscles of a limb', recommend being explicit about the intratumoral nature of the treatment. Further, it would be informative for the reader to add treatment interval in this section and modify the heading of the section to 'Treatment and Safety'.

Response: the number of treatments are in the results section as requested. We have clearly specified the intratumoral injection procedure in the Methods "Procedures" and it is also mentioned in the abstract.

Comment 7

Page 5, 'eight AEs due to other attributions led to LOAd703 treatment discontinuation', please add more details.

Response: This sentence was removed in sake of shortening the manuscript but we refer to the AE and SAE tables in which adverse reactions due to any cause are shown as in the paper, the AEs attributed to LOAd703 are shown.

Comment 8

Page 6, Immune biomarker analyses, 'To evaluate the capacity of LOAd703 to stimulate a Th1 type of inflammation in the TME'. Atezolizumab should be added to the sentence as the separate impact of LOAd703 cannot be assessed in this trial.

Page 6, the authors report several TME changes based on expression analyses. ' The data confirm an upregulated gene expression of Th1 biomarkers including markers indicating the presence of T and NK cells, their activation and cytotoxic function, chemokines attracting T and NK cells to the TME as

well as receptors that allow them to transmigrate from blood vessels into the TME. Further, markers for antigen presentation and B cell activation were upregulated.' More details with indication of specific markers should be added here in Results instead of in the Discussion.

Page 6, Significant differences in gene expression level between the two groups are displayed for these groups in heatmaps and volcano plots at baseline. Please revise the sentence and remove 'for these groups'

Response: In sake of space, the biomarker description was needed to be shortened, however, in the supplementary tables full lists of all significant makers are shown. It also made more sense to discuss the markers in the discussion but **if the Editors of Nature Communication agree, we would be happy to provide a longer biomarker section in the results?**

Comment 9

Page 7, please add information on PR and CR rate in addition to ORR.

Response: as the reviewer wanted us to not describe the results too much, this was a somewhat contradictory comment (see Comment 3). However, it is mentioned in the section of patients with particular benefit from treatment.

Comment 10

Page 8, Patients with particular benefit from treatment: 'we observed two patients with partial response (PR) who became tumor-free after study participation'. This is a tricky formulation as one patient had a residual lesion removed and apparently the other was in complete metabolic response. Was a lesion still visible on CT ?

Page 8, Discussion. It is recommended to remove this unsubstantiated sentence; 'The survival appeared clearly superior to what is expected without treatment.'

Page 8, 'ICI targets were simultaneously upregulated in injected metastases, including PD-L2 [PDCD1LG2], LAG-3, and TIGIT which indicates a benefit from combining LOAd703 with other checkpoints than atezolizumab'. Recommend revising to 'potential benefit'

Page 9, top sentence on Opdivo please insert reference on phase III trial.

Page 9, 'Patients who survived longer had high levels of genes related to....'. Please take into consideration in the discussion potential differences in prognostic characteristics of the patients.

Page 9-10, 'The results seemed clearly superior compared to chemotherapy or best supportive care, which would have been the alternative options for the vast majority of the study participants...' Some of the BRAF mutated patients did not receive BRAF/MEK inhibitor and ipi/nivo therapy before inclusion. Please take this into consideration.

Response: The discussion is overall shorter and removed sections that troubled the referee.

Comment 11

Page 20, 'Patients with mucosal, acral or uveal melanoma were not included in this study due to the differences in biology and therapeutic response from CMM.' According to the protocol in sept 2020 'Exclusion criteria no 1 was modified so that patients with mucosal melanoma will no longer be excluded'

Page 21, procedures: A definition of metabolic response is lacking.

Table 1, important information is lacking on PDL1 expression, M1 stage, and grading of LDH level

Table 2, Injection site nomenclature is unclear, ' Abdomen, Anterior chest, Axilla, Leg (gluteus, thigh, knee, lower)', please add info on whether it is sc, LN or muscular lesions.

Response: The methods section has been modified to correctly describe the exclusion criteria accordingly to the last protocol and actual enrolled patients. The metabolic response is described in the Supplementary Table 8 footnotes. Patients with elevated LDH at baseline and M1 stages are mentioned in Table 1 but unfortunately, information about PDL1 expression by histochemistry is not available. Likewise, the collected data on injected tumor site does not provide additional information.

Reviewer #2 (Remarks to the Author):

In this article, the authors report the results of a phase I/II trial of atezolizumab IV plus IT administration of a viral vector encoding trimerized, membrane-bound CD40L and 4-1BBL. Therapy was generally well tolerated, with adverse events expected from IT immunotherapy and PD-L1 immune checkpoint inhibition (ICI). Biomarker data suggest clear activation of an immune response. While the ORR is relatively low, OS data seem more promising, especially in this heavily pretreated population.

LOAd703 is novel intratumoral viral therapy which encodes and transfers Th1 type immunostimulatory genes to the TME. While combinations of ICI with oncolytic viruses have been investigated before in phase 1-3 studies (e.g. T-VEC + pembrolizumab, MASTERKEY-265/KEYNOTE-034 study, Ribas et al. ESMO 2021), this is - to the best of my knowledge - the first early-phase trial investigating this specific virotherapy in combination with PD-L1 ICI in PD-1 ICI-refractory melanoma.

I don't have any remarks on the study design and statistical design. The trial is well executed. The manuscript is well written. Data are qualitative, well presented and informative. The conducted exploratory analyses on blood and biopsies are relevant, informative and provide a biological surrogate for drug activity. Furthermore, they seem to identify a population who derives more benefit from this therapeutic approach.

On a side note, it is not clear why not all patients underwent biopsies at baseline and during treatment.

I have some remarks:

Comment 1

- The authors present data showing that the lower dose of LOAd703 seems associated with better OS. I can acknowledge that the number of patients is very low, but I miss some critical insight into this observation. Could this for example be due to more immune exhaustion after repetitive exposure to viral antigens? Or because the median number of injections was lower in the high dose group?

Response: In this study we cannot say why the lower dose patients seem to survive longer than the higher dose. As the referee comment, and we discuss, the number of patients are low. However, taken all LOAd703 studies together (LOKON001, 002 and 003) we always see the same trend and we agree that repetitive treatment with a higher viral antigen load may be responsible but thus far there

is no proof to discuss. The number of injections will be lower if the patient progress and go out of study but that is likely more circumstantial as that patient population progress for another reason earlier than those that survive longer.

Comment 2

- It is unclear whether atezolizumab has an additive/synergistic effect in this combination therapy in this PD-(L)1 ICI refractory population or whether the observed effect is due to LOAd703 alone (see also effect of T-VEC monotherapy for example OPTIm trial). Could the authors comment on this?

Response: Of course, in a pivotal study the suitable control would be to have each agent alone and in combination. However, as the patients are refractory to one, or several rounds of a PD-1 antibody alone or in combination with other antibodies or even Tvec, continuing to treat these patients with a PD-1/PD-L1 targeting antibody without any other additions was not ethically appealing. However, the virus alone can be used in a control group moving forward. Nonetheless, Tvec has not gain a lot of interest to be used as a monotherapy and the combination is likely a better option for the patients. Our in vitro experiments and animal data supports the combination to a greater immune activation and in animals the combination supports the systemic response post intratumoral injection of LOAd703 which is needed in this patient group with multiple metastases.

Comment 3

- I miss some insights on future perspectives with this novel therapy. The authors state that investigating nivolumab-relatlimab +/- LOAd703 in the first-line setting would be attractive, presumably due to upregulation of LAG-3 shown in this study. However, one could also combine CTLA-4 ICI based on these data (also based on the fact that CTLA-4 may have an important involvement in the TME).

Response: It is still debated how to move forward in melanoma with LOAd703 and there are no formal decisions made yet. Any or several ICIs may be of interest. The study PIs advice the company currently sponsoring the clinical development of possibilities.

Comment 4

- Multiple immune resistance mechanisms have been described, especially in the later-line setting. The exploratory data shown in this study indicate that baseline expression of genes involved in antigen presentation and protection against a viral response at baseline appears to be associated with better OS. This could serve as a basis for future biomarker research to select patients for whom this therapy could be of particular benefit. Could the authors comment on this?

Response: This is correct, we are currently analyzing all biomarker data with focus on the patients with progressive disease and short survival to determine if they have a joint biomarker signature. Preliminary, we have defined a patient population within our study based on biomarkers in which none of the patients survived more than 6 months (median OS 4 months). Analyses are ongoing and a paper will be prepared within the next few months.

Comment 5

Minor improvements:

- Minor spell check required, for example:

* Introduction:

** "has revolutionized...and is standard-of-care" instead of "have revolutionized...and are"

** "may depend" instead of "may depends"

- Table 3:

* Please review pyrexia AE: 14 G1-2 and 1 G3, but 14 any grade AE.

Response: We have completed a correction of the manuscript and in regards to the last comment, the same patient can have more than one pyrexia and it may be of different grades separated in time. Hence, a single patient can be in both G1-2 and G3 group so end number of affected patients is still 14.

REVIEWER COMMENTS

Reviewer #1 (Remarks to the Author):

The authors have to some extent addressed my comments including adding information on M1 stages which points towards marked differences in subgroups among the two patient cohorts. Thus, the low dose cohort includes only 14 % M1c patients while the high dose has 41% M1c. The authors have added text and a figure on survival according to M1 stage in the result section. However, the total number of patients as well as the numbers of patients in the different M1 stages do not harmonize between table and figure/figure legend. Table: M1b: n=11, M1b: n=5, M1c n=:8, with a total of 24 patients. Figure 3H: M1b: n=12, M1b: n=2, M1c: n=16 with a total of 30 patients. Figure 3H legend: M1c, n=12; M1c, n=2; M1c, n=10 with a total of 24 patients. If the numbers given in the table is to be trusted then new calculations of survival in Figure 3H needs to be carried out and associated text revised.

Further, the authors have now included a cox regression analysis claiming that “ no clinical parameters impacted on survival”. However, from the related suppl tabl S9 it appears that M1 stage is not included. M1 staging is one of the strongest prognostic indicators for survival and should be included in the cox analysis.

The impact of M1 stage in regard to interpretation of results between VP dose cohorts in the trial, and in comparison, with other trials, have not been integrated in the Discussion. It is recommended to incorporate a paragraph on trial limitations highlighting this together with the potential impact of post PD treatment on OS as previously mentioned.

Response: Thank for observing the mistake in Table 1. We have revised and performed QC checks for the M1 data which is now revised. We also added M1 staging in the Cox analysis in Supplementary Table S9 and revised the manuscript accordingly on pages 6 (results) and 8 (discussion).

Reviewer #2 (Remarks to the Author):

I thank the authors for their comprehensive replies.

In the discussion, I still miss some future perspectives with this treatment strategy. Should this be investigated in earlier treatment lines or in combination with other drugs? Could this be a biomarker-driven drug and if so which one? The authors should scientifically comment on this regardless on strategic decisions of the biotech company producing the drug.

Furthermore, I have some small spelling suggestions:

INTRODUCTION:

Immune checkpoint inhibition (ICI) targeting PD-1, CTLA-4, and LAG-3, HAS

revolutionized ... and IS standard-of-care together with protein kinase inhibitors for BRAFV600E-mutated CMM

DISCUSSION:

additional ICI than atezolizumab

Response: Thank you for noticing the spelling errors, we have modified them accordingly to the suggestions. We also agree that it may be of interest for the readers to have a more elaborate future perspective discussion, which we have added on pages 7 and 8.

Reviewer #3 - Biostatistics, clinical trials (Remarks to the Author):

The paper reports results from a Phase I/II trial that employed a BOIN design for dose escalation across two dose levels. The trial planned to enroll 25 evaluable subjects at the MTD (up to 50 total), but was terminated early due to lack of efficacy signal. Dose escalation occurred after seven patients were enrolled at the initial dose level. Given the small sample size (N = 24), all analyses are exploratory and should be interpreted with caution, as noted by the authors. Nevertheless, the statistical reporting requires greater rigor and improvement. Specific recommendations include:

1. The rationale for N = 25 appears to be based on efficacy assumptions. However, the statement lacks key details, such as the assumed true response rate under the alternative hypothesis. The study protocol includes this information; authors should reference the protocol as supplementary material or cite the published protocol if available.

Response: We thank the reviewer for the statistical review. We agree that a reference to the protocol is suitable for this purpose and have added that in the manuscript study design on page 18.

2. All point estimates (e.g., ORR, PR, SD rates, median PFS/OS) should be accompanied by 95% confidence intervals, as specified in the protocol.

Response: We have now included the confidence interval for all point estimates in Table 2.

3. Statistical methods used for all analyses should be described in the main text. For example, methods mentioned in figure titles (Figures 1–2) should be expanded with sufficient detail.

Response: We have now added the statistical methods not only in the Figure Legends but also in the method section at page 20.

4. AE table (Table 3) should present adverse events by dose level, consistent with the study SAP.

Response: Table 3 is modified to contain AEs by dose level.

5. Figure 1A compares anti-adenoviral antibody levels over time by dose level. The use of two-way ANOVA for longitudinal measurements is not appropriate; a repeated-measures or mixed-effects model would be more suitable.

Response: We agree with the reviewer and has changed the statistics to a mixed-effects analysis.

6. Figure 1B correlates antibody units at week nine with overall survival using simple linear regression, which is inappropriate given the presence of censored survival data. A Cox proportional hazards model or similar approach should be considered.

Response: We agree that under normal circumstances, censored patients will disable the linear regression analysis. However, the study had an unusually long follow-up for survival, and the censored patients were all surviving >17 months (n=1), >28 months (n=2) or >30 months (n=7), and none of them were among the three patients that had higher anti-adenoviral antibodies. Hence, if the actual death date were later for the censored patients, the week 9 level of anti-adenovirus antibody would still be the same, and in the same range of most patients dying early. The regression plot visualizes the three patients with higher level of anti-adenovirus versus the remaining patients that all had a median level of antibody independently of short or long survival in a manner that makes the figure relevant. However, to avoid any confusion or misunderstandings, we have explained the censored patients in the figure legend, in the results (page 5) and discussion (page 8), and renamed the x-axis from overall survival to last known survival date.

7. It is unclear why some analyses in Figure 2 include multiple comparison adjustments while others do not. Adjustment should be applied consistently across marker comparisons.

Response: This has now been clarified in the results section page 5 and 6 and discussed on page 8 to make sure the reader can see that when dividing the data sets they become too small for multi comparison compensation which increases the risk of false negative data. Hence, these sub-cohorts were also analyzed without multi comparison to evaluate if there are possible masked biomarkers that are still relevant to evaluate in upcoming studies with larger cohorts or with single-plex methods. It is a balance to have too harsh or too poor statistics as false negatives as well as false positives can lead to wrong conclusions. We agree with the reviewer that it is important that the reader understands what data is presented and the results section is improved in this perspective.

8. Information on study drug compliance is missing and should be included.

Response: The study drugs LOAd703 and Atezolizumab are administered at the hospital as an image-guided intratumoral injection versus intravenous infusion, respectively. Hence, the exact number of doses per patient is registered in the study records and summarized in Table 2.

Minor Comments

1. Phrases such as “significant difference at a 95% confidence interval” are unconventional. Recommend revising to “at a significance level of 0.05.”

Response: We have changed it accordingly to the suggestion from the reviewer.

2. In the Results section (page 4, lines 85–89), some numbers cited in the text do not appear in Table 1.

Response: We have now added the missing information in Table 1.

3. In the Methods section titled “Statistical analyses for clinical assessment” (page 19, line 522), the term “one-sided binominal t-test” is unclear. For a binary endpoint like ORR, a one-sided binomial test would be appropriate, but not a t-test.

Response: That is correct, and we have corrected the error in the Method section.

Reviewer #4 (Remarks to the Author):

Response: We acknowledge that Reviewer 4 co-reviewed this manuscript and did not have further separate questions or comments to us.

REVIEWERS' COMMENTS

Reviewer #1 (Remarks to the Author):

I have no further comments

Reviewer #3 (Remarks to the Author):

We thank authors for fully addressing almost all of our prior comments. The only revision needed is about the reported confidence intervals. By convention, the confidence interval is written as (lower boundary, upper boundary) or lower boundary-upper boundary. But some CIs currently have upper boundary reported first followed by lower boundaries. These should be corrected.

Response: This has now been corrected in Table 2

Reviewer #4 (Remarks to the Author):
